# What Drives Land Use Change in the Southern U.S.? A Case Study of Alabama

**Eugene Adjei [1], Wenying Li [2], Lana Narine [1] and Yaoqi Zhang [1,\*]**

[1] College of Forestry and Wildlife Sciences, Auburn University, Auburn, AL 36849, USA
[2] Agricultural Economics and Rural Sociology, College of Agriculture, Auburn University, Auburn, AL 36849, USA
\* Correspondence: zhangy3@auburn.edu

**Abstract:** Land use change reflects fundamental transformations in society. To better understand factors contributing to current land use changes in Alabama, we expand on existing land use studies by employing a generalized least-square method nested in a system of equations for the analysis. We correct for endogeneity issues in our paper by incorporating a control function technique. Using repeated land use data from 1990–2018, we focus on analyzing factors affecting land use changes among timberland, agricultural, urban, and conservation land use types. Our results reveal that land quality factors influence land allocation and land use decisions. We also indicate that population density is a driver for replacing timberland for urban development and agricultural purposes. We show that interest rates are important factors in timberland use decisions as timberland investments are sensitive to capital cost. We provide a basis for future simulations of nationwide land use changes under different economic and policy scenarios, as we offer new insights and contribute to the existing knowledge into public policies that are related to land use planning and management.

**Keywords:** land use change; land quality; interest rate; commodity prices; timberland; agricultural land

## 1. Introduction

Land use change is an important sustainability issue in the world. These changes occur due to human and economic activities that often lead to transformation in the ecosystem. Understanding land use changes and their potential drivers are critical for designing policies to address issues related to food security, wildlife management, urban growth management, and climate change to ensure global environmental sustainability. We address land use changes in the U.S. south, specifically Alabama, by focusing on land use decisions of private landowners. Private landowners account for about 60% of land ownership in the U.S., including 58% of the country's timber resources, and 90% of the nation's agricultural lands [1]. Thus, private landowners are fundamental to the concerns over the social costs and benefits associated with land use changes. Subsequently, individual land use decisions generate social costs, and benefits are not reflected in land rents leading to externalities [2]. These pervasive externalities provide great opportunities to develop policies that focus on aligning private land use decisions with social objectives to meet both local and global sustainability goals.

To develop all-inclusive policies to address land use, we determine the drivers of land use transformations in the U.S. south, specifically Alabama, in three principal ways. First, and most importantly, we consider natural and economic factors that influence all the major land use types. Natural factors (hurricanes, storms, extreme weather variation), and investment factors (interest rates) are determinants that are ignored in past studies. Therefore, we determine how the recently updated list of disaster-prone counties influences private land use decisions. Second, we employ an econometric approach that accounts for correlation among the various land use types, resolves omitted variable bias, and reverse

causality. Third, in contrast to other studies that conduct large-scale land use changes, we focus on Alabama to prevent the loss of significant local trends and variations to develop specific land use policies that address current land use challenges in the region.

Alabama accounts for a number of land use changes in the south of the U.S. This is due to modifications in various land use policies [1]. These modifications have implications and long-lasting effects on private landowners. For instance, if new land reforms affect land revenue, then private landowners are likely to reconsider their decisions and will shift to other land use types to maximize revenue. Subsequently, Alabama is experiencing exponential growth in infrastructure, transportation systems, and a change from agrarian land use to timberland and urban land use [2]. Thus, farmlands that are closer to growing urban areas have appreciated, making it difficult to access land for agriculture. Again, there are frequent occurrences of extreme weather conditions that have caused damage to properties in the region. The United States Department of Agriculture (USDA) has updated the list of disaster-prone counties in the region due to extreme weather situations. We consider these new events in our analysis as they are essential and have consequences for future land use decisions. Lastly, we achieve our objectives by using an extensive data collection method to construct county-level estimates to explain current land use decisions by private landowners. This study is a true reflection of land use decisions due to similarities in land use practices, land ownership structures, and land management strategies in the U.S. south [3]. Therefore, our findings provide relevant information for policy decisions to meet potential land use demands [4].

Approximately 90% of the total land area in Alabama is rural and has undergone numerous changes over the years [5]. For instance, agricultural lands have decreased by about 27% while timberlands and urban lands have increased by about 8% and 13% respectively from 1972 to 2000 [6]. Within the timbered landscape, hardwoods constituted the highest in the 1970s; however, recent timberland outlooks show that softwoods have increased relative to hardwoods [7]. Future projections reveal more land use changes that are likely to favor urban and timberland use [8]. The projections show that urban lands are usually allocated for transportation, infrastructure, and housing purposes while timberlands favor softwood establishments and partly recreational activities [6]. These land transformations emanate from social, economic, and political processes that are embedded in society, as the above-mentioned factors influence the demand for, and supply of, land for land-related resources [4,5]. It is imperative to compute the land use dynamics and ascertain how different land use types respond to socio-economic pressures, various market structures, natural factors, and investment. We expect this study to provide information on the drivers of land use changes in Alabama and other post-industrial countries.

Past studies have tried to identify factors of land use change in the contiguous U.S. [8–14]. Other studies have been conducted to evaluate the causes of forest degradation, deforestation, and agricultural expansion in developing countries [15–19]. In the abovementioned studies, land use theories and allocation models are adopted as the theoretical frameworks. These frameworks serve as the basis to develop hypotheses and research objectives. To explain land use changes in the U.S., county-level data [8–11] and inventory data from the National Resource Inventory(NRI) are employed [12–14]. Due to the lack of data, researchers use satellite imagery and other remote sensing techniques to acquire data to analyze land use changes in developing countries [15–19]. The NRI and satellite approaches rely on pixels, parcels, or sample points to explore land use change [10,14,20–25]. County-level data is typical to the U.S. and is aggregated land use data collected over years [8,9,11–13].

Explanatory variables adopted to explain land use changes are a function of proxies of land rents from own use, alternative use, land quality, and other economic indicators. These variables can be spatial, economic, and demographic factors, such as prices, population, income, land slope, precipitation, temperature, and natural disasters [8–19]. Land use determinants operate as proxies for land use change and vary across studies. However, most socioeconomic indicators often utilized are net revenue, land value, com-



modity prices, and government subsidies [6,8,10–12]. Environmental factors that serve as a proxy to explain land use changes include weather conditions, storms, hurricanes, and drought [20–22]. Studies show that net revenue is important for choosing a particular land use type. For instance, earlier studies adopted net returns from crops, timber, and pasture as explanatory variables to assess major land use changes in the southeast U.S. [6,8–11]. Other studies rely on net returns to explain land use decisions, especially between agriculture and forest [16,18]. Generally, the conclusions from the abovementioned studies suggest that landowners allocate a piece of land to the use that generates the highest net return. Also, the nature of the land has implications on landowner's land allocation decisions. Thus, researchers consider slope, soil quality, and water-holding capacity to explore land allocation decisions. The results indicate that landowners allocate land with high quality and good water-holding capacity for agriculture while land with poor quality is usually allocated for timber production [10,13]. Although land quality indicators are not included as explanatory variables, anecdotal evidence suggests that the conversion of natural forests for agriculture in developing countries is due to the fertility and moist nature of forest soils. Socioeconomic indicators affect land allocation decisions. For example, studies [6,8,15] show that population density affects timberlands negatively, but affects agricultural and urban lands positively, as an increase in the number of people puts pressure on natural resources and demands additional land for food production. These findings align with a study that suggests that population growth leads to deforestation and forest degradation in developing economies [17].

Estimates of landowner's response to economic returns are important due to increased attention to economic incentive-based land-use policies. We improve on existing literature by including conservation lands in our analysis. Conservation lands are critical, as the government compensates private landowners that enroll their lands for environmental benefits, such as soil erosion control, biodiversity and wildlife improvement, and agricultural price commodity controls [26]. These payments are incentives and also contribute to land use change, as money for tree planting is used to defray timberland investment costs [10,14]. It is essential to establish that the CRP rental payments subsidize about 22% of total private tree planting costs in the U.S. south [27]. In this paper, we adopted a control function method to correct the endogeneity problems. To the best of our knowledge, no study has applied this method to correct endogeneity in land use studies. A closely related study by [18] uses a 3-stage least-square (3SLS) to account for the endogeneity between drivers of deforestation and forest degradation in Vietnam. However, we prefer the control function approach, as it is flexible and accommodates linear and nonlinear relationships among the variables [18]. Subsequently, we utilize a panel seemingly unrelated regression (PSUR) nested in a generalized least square for the econometric analysis. Past studies [28,29] adopted a seemingly unrelated regression method to estimate land use changes. However, this method only assumes the correlations among the error terms across the different land use types without accounting time-invariant effects. Thus, we employ the PSUR method that accounts for both time-invariant effects and also takes into consideration the correlation across the land-use error terms [30].

We construct a balanced panel dataset that covers time-series and cross-sectional observations. Subsequently, we include interest rate as an explanatory variable in our analysis to capture the effect of capital cost on land use. Thus, we establish how interest rates affect investment decisions in the land use types. We also consider how natural disasters affect land use by including the list of disaster-prone counties in our analysis.

## 2. Materials and Methods

### 2.1. Analytical Framework

We discuss the analytical framework that describes factors of land use changes in this section. The Ricardian rent and the Von Thünen theories are the most extensively utilized models in land use literature. These models can be transformed into acreage allocation models that relate land use shares to factors of land rent [13]. We recognize

that land use changes are the net result of flows between the different land use categories and, as the number of land use alternatives increases, the amount of land transformation increases. Therefore, we use repeated land use information to model land use changes over time [20]. In addition, understanding the pattern of flow is important for policy formulation. For instance, the conversion of agricultural land for urban use may be of concern to policy-makers, while the conversion of agricultural land for timber purposes may not be of great concern from the perspective of maintaining open space. Similarly, newly planted timberlands have different ecological characteristics than an old-growth forest, with implications for the provision of environmental benefits.

The theoretical basis for our study is land rent maximization. Land rent is a residual economic surplus, where total revenue accrued from land use exceeds total cost [31]. Land use theories are built on previous knowledge, defining land rent as "that portion of the produce of the earth that is paid to the landlord" [32]. We extend the idea into this paper to incorporate transportation and land costs, where the model explains that the price of land increases as one moves closer to a city or business center [33]. We hypothesize that with a fixed land unit, the relative land rents are determined by allocating among competing uses, and that a neutral landowner allocates a piece of land to the use that maximizes the overall net expected return. We assume heterogeneity in the land use types by considering land quality differences. We also assume simultaneous use of lands in the presence of the various land use types as we seek to determine how socioeconomic and natural factors affect land use decisions. These factors are significant, as they influence land allocation for a specific use. Although land use decisions are made by private individuals, these decisions occur within a framework that is often affected by public investments and incentives.

We evaluate four major land use types (timberlands, croplands, urban lands, and conservation reserve program (CRP) lands) with a system of equations. We transform the land acreages into shares to derive elasticities that relate changes in land use to changes in land use determinants. The elasticities reflect the response of land use changes to changes in the various determinants. Land use is dynamic and responds to changes in determinants differently. For instance, urban land use is more sensitive to income and demographic characteristics compared with other land use types [34]. Also, land allocation is affected by land quality, which affects land use for various uses [11]. For instance, fertile land is very important for agricultural use but less important for urban land and timberland use. Factors that affect timberland are more complicated, due to its multiple uses for timber production and recreation. High income might be associated with higher labor costs that may negatively affect more labor-intensive production, such as agriculture, but have a positive effect on the need for recreation and leisure purposes that are associated with timberland. Urban land use is very different from agricultural and timberland use, though it might be more associated with population density and land parcelization [35].

### 2.2. Empirical Strategy

To estimate the factors that affect the various land use categories, we first account for endogeneity by using the control function method. The control function technique is a two-stage residual inclusion (2SRI) strategy that mitigate endogeneity by estimating an auxiliary regression in the first stage to generate residuals for the second stage analysis [36]. In the second stage, we include the residuals from the first stage as additional regressors together with the endogenous variable. In this instance, the residuals mitigate the endogeneity in the regression, as they serve as proxies for the factors in the error term that are correlated with the endogenous variables [37]. Another concern is the reverse causality between the land types and their respective prices. We adopted the Hausman-type instrumental variable (IV) method to resolve this problem. The idea here is that prices from other counties are correlated and can serve as an IV for a particular county. Using neighbors' prices to instrument product prices are widely accepted in the industrial organization literature [38,39]. We specify the first stage equation as $ln(P_{it}) = f(ln(P_{jt}), v_i)$ where $P_{it}$ is the vector of prices from the land categories (timberland and agricultural land) in county

$i$ at time $t$; $P_{jt}$ denotes the vector of prices from each of the land categories in county $j$ at time $t$, excluding the county $i$. County $i$ is excluded from the construction of instruments to reduce simultaneity bias caused by common county-specific enrollment shocks. We also control for county-specific fixed effects using $v_i$. We correct for the simultaneous causality for population density and income per capita on urban land, as counties with higher income and population density may attract more urban lands. In addition, urban sprawl may change the nature of agriculture. For example, increasing urban lands may alter agricultural lands from land-intensive crop production to labor-intensive recreational agriculture. Since it is difficult to find explicit IVs for such challenges, we adopt the lagged values of the endogenous variables as IVs for our analysis. We also apply the control function method here. In dealing with omitted variable bias, we introduce individual-and-time fixed effects and a dummy variable to account for extreme climate and weather conditions to eliminate the predictability of natural disasters [22]. The dummy variables denote counties that are disaster-prone and disaster-free.

We utilize the multistep generalized least-square (GLS) method to estimate the parameters of the PSUR model, based on the assumption that the land use types are correlated. Since all the lands are in concurrent use, it indicates that there are correlation issues that require consideration. Thus, we employ the PSUR method that accounts for the correlation. We set up the private landowner's land allocation model by the following system of equations:

$$
\begin{aligned}
ln(Timber_{i,t}) = \quad & \beta_{1,1}ln(popdensity_{i,t}) + \beta_{1,2}ln(capita_{i,t}) + \beta_{1,3}ln(timprice_{i,t}) \\
& + \beta_{1,4}ln(land\ quality_{i,t}) + \beta_{1,5}ln(crop\ price_{i,t}) \\
& + \beta_{1,6}interest\ rate_t + \beta_{1,7}disaster_{i,t} + \pi_{1,9}\hat{\epsilon}_{i,t} + v_{1,\ i} + \mu_{1,i,t}
\end{aligned}
\tag{1}
$$

$$
\begin{aligned}
ln(Agric_{i,t}) = \quad & \beta_{2,1}ln(popdensity_{i,t}) + \beta_{2,2}ln(capita_{i,t}) + \beta_{2,3}ln(timprice_{i,t}) \\
& + \beta_{2,4}ln(land\ quality_{i,t}) + \beta_{2,5}ln(crop\ price_{i,t}) \\
& + \beta_{2,6}interest\ rate_t + \beta_{2,7}disaster_{i,t} + \pi_{2,9}\hat{\epsilon}_{i,t} + v_{2,i} + \mu_{2,i,t}
\end{aligned}
\tag{2}
$$

$$
\begin{aligned}
ln(Urban_{i,t}) = \quad & \beta_{3,1}ln(popdensity_{i,t}) + \beta_{3,2}ln(capita_{i,t}) \\
& + \beta_{3,3}ln(timprice_{i,t}) + \beta_{3,4}ln(land\ quality_{i,t}) \\
& + \beta_{3,5}ln(crop\ price_{i,t}) + \beta_{3,6}interest\ rate_t + \beta_{3,7}disaster_{i,t} \\
& + \pi_{3,9}\hat{\epsilon}_{it} + v_{3,i} + \mu_{3,i,t}
\end{aligned}
\tag{3}
$$

$$
\begin{aligned}
ln(CRP_{i,t}) = \quad & \beta_{4,1}ln(popdensity_{i,t}) + \beta_{4,2}ln(capita_{i,t}) + \beta_{4,3}ln(timprice_{i,t}) \\
& + \beta_{4,4}ln(land\ quality_{i,t}) + \beta_{4,5}ln(crop\ price_{i,t}) \\
& + \beta_{4,6}interest\ rate_t + \beta_{3,7}disaster_{i,t} + \pi_{4,9}\hat{\epsilon}_{i,t} + v_{4,i} + \mu_{4,i,t}
\end{aligned}
\tag{4}
$$

where $ln(Timber_{it})$, $ln(Agric_{it})$, $ln(Urban_{it})$, and $ln(CRP_{i,t})$ denote the logarithm of timberlands, agricultural lands, urban lands, and CRP lands in county $i$ at year $t$. $v_i$ denote the unobserved individual–level effects; $\mu_{i,t}$ signify the observation-specific errors; $\hat{\epsilon}_{it}$ is a vector of residuals we generated from the first stage regression and plugged into the system of equations to eliminate the endogeneity issues. The residuals are commodity prices (timber and crop prices), population density, and median household income per capita. The explanatory variables $ln(popdensity_{i,t})$, $ln(capita_{i,t})$, $ln(timprice_{i,t})$, $ln(land\ quality_{i,t})$, $ln(crop\ price_{i,t})$, and $interest\ rate_t$ are logarithms of population density, real median household income per capita, real timber price, land quality, crop price index, and interest rate, respectively. The explanatory variable, $disaster_{i,t}$, is assigned 0 or 1, where 0 denotes counties that are not disaster-prone, while 1 denotes counties that are disaster-prone. The dummy variable has a panel data dimension, as certain counties that were not disaster-prone are now prone to natural disasters. The interest rate is subscript $t$, as it is fixed for each county and only varies over time. All explanatory variables are in logarithm terms except interest rate, since it is already expressed as a percentage.

*2.3. Data and Land Use Statistics*

We group the land use types into four categories (timberlands, agricultural lands, urban lands, and CRP lands). We exclude public lands, transportation infrastructure, and miscellaneous lands (water bodies and barren lands) from the land use types as changes in these land types are not reflective of utility maximization of landowners. Again, miscellaneous lands are difficult to quantify and estimate due to the unavailability of net return to use. Timberlands consist of hardwoods, softwoods, and mixed hardwoods. We obtained the timberland data from the Forest and Inventory Analysis (FIA) database for 1990, 2000, and 2018. The FIA is a panel survey consisting of timber and other timber-related data that are collected across private timberlands. Timberland data for 2010 were unavailable, so we applied a linear interpolation method to generate observations for 2010 based on the other observed timberland data points. Agricultural lands comprise irrigated croplands, non-irrigated croplands, and pastureland. We obtained data on agricultural lands from the Agricultural Census of the United States Department of Agriculture (USDA-NASS) Census of Agriculture database for 1992, 2002, 2012, and 2017. The USDA-NASS database is a comprehensive tool for accessing published data on agriculture. Urban lands comprise residential lands, industrial areas, and other developed areas. The data on urban lands were obtained from Bureau of Census reports. As timberland areas fluctuate less frequently relative to agricultural and urban lands, we interpolate the data series to produce a consistent set of observations for our analysis. We obtained data on CRP lands from the Farm Service Agency (FSA) of the USDA. CRP lands are sensitive and highly erodible agricultural lands that are enrolled in conservation and other environmental programs. The FSA is an organization that implements agricultural policies, administers credit policies, manages conservation programs, controls commodity prices, administers disaster programs, and implements farm marketing projects.

It is a fact that commodity prices cause land use changes [8–10]. Thus, we account for the effect of prices on the various land use types. We include timber prices to determine their effect and their relative effect on other land use types. We compute timber prices using stumpage prices for different timber species that we collected from the TimberMart-South quarterly reports. TimberMart–South is an organization that reports on current and long-term trend data on stumpage and delivered wood prices and other fundamental forestry business information. We deflated the timber prices using the producer price index for all commodities. Likewise, we constructed a Laspeyres crop price index to determine its effect on agricultural lands and the other land use types. A county-level crop production and state-level crop prices are collected from the USDA-NASS quick stat website to construct the crop price index. We use the 8 major agricultural commodities that are produced in the U.S. Following [40], we compute the crop price index for the 8 major crops (corn, wheat, cotton, oats, peanuts, soybeans, sorghum, and barley) as follow:

$P_{it}^{a} = \left( \sum_{l=1}^{8} Pl_t Ql_{i1990} \right) / \left( \sum_{l=1}^{8} Pl_{t1990} Ql_{i1990} \right)$, where $Pl_t$ is the deflated price received for crop $l$ at time $t$; $Ql_{it1990}$ is the crop production of crop $l$ in county $i$ at time 1990 as the base year.

We include population density as an explanatory variable to account for the effect of population dynamics on land use change. Population growth is a potential driver that induces the establishment of developed areas. Thus, we determine how the growing population contributes to land use change. We define population density as: *Population Density*$_{it}$ = (*Population*$_{i,t}$/*Land area*$_i$), where *population*$_{i,t}$ is the number of people in county $i$ at time $t$. *Land area*$_i$ is the total land area in each county. The data on population and land area were obtained from the Bureau of Economic Analysis (BEA) and USDA respectively. The BEA is a government agency that provides data on macroeconomic and industry statistics for the nation and various states within the nation. To account for how household income per capita affects land use changes, we obtained county-level median household income per capita data from BEA. Although the effect of household income has relative and ambiguous effects on various land use types, past studies indicate that household income tends to affect timberland through recreation and drive the demand

of land for urbanization [41]. Thus, we include household income to determine its effect on land use. The median household per capita income data is deflated using the consumer price index to account for inflation. Additionally, we include interest rate in our analysis to determine how access to capital and macroeconomic factors impact land use. Interest rates are an important determinant of land allocation, as they tend to influence the cost of borrowing, the return on savings, and determines the total return on land investments. We obtained interest rates data from the Federal Reserve Economic Data (FRED) of St. Louis database.

We include a dummy variable to capture natural factors and extreme weather conditions. This is important, as the disaster-designated counties and their associated beneficiaries keep increasing over time. Initially, disaster-designated counties were usually located in District 2, District 3, and District 5.. However, the number of counties in the disaster-designated areas has expanded due to frequent storms and hurricanes [42]. We include land quality in our analysis. These land quality classes represent various land characteristics. We calculated land quality as a weighted average of land acres in each land class for each county based on the soil types. The NRI classifies the land quality from I to VIII, where I is the most productive soil and VIII is the least productive soil. The land qualities are categorized into three classes, namely: land quality I-II, land quality III-IV, and land quality V-VIII. Land quality is a critical indicator of land use changes as different soil characteristics influence land productivity.

Table 1 presents the summary statistics for the different land use types and explanatory variables for the study. We assume that land conversion costs are implicitly captured by the net returns of land use and the land quality. Thus, we proxy cost with the land quality characteristics, as data on cost are unavailable for this study. We assume that low quality lands with poor and unfavorable characteristics tend to increase in conversion cost compared to lands with high quality and favorable characteristics.

**Table 1.** Descriptive Statistics (268 observations: 67 counties for 4 time periods).

| Variable | Mean | Standard Deviation |
|---|---|---|
| Dependent variables | | |
| Timberlands | 0.69 | 0.15 |
| Agriculture lands | 0.28 | 0.14 |
| Urban lands | 0.02 | 0.01 |
| CRP lands | 0.01 | 0.02 |
| Independent variables | | |
| Population density (Number of people/acre) | 133.49 | 156.05 |
| Income per capita (Dollar ($)/number of people) | 12,842.41 | 2458.55 |
| Interest rate | 4.04 | 2.93 |
| Land quality I-II | 0.22 | 0.12 |
| Land quality III-IV | 0.34 | 0.13 |
| Land quality V-VIII | 0.43 | 0.17 |
| Timber Price ($/m$^3$) | 0.39 | 0.15 |
| Crop price Index | 335.7 | 17.01 |

We compute district-level land use changes by grouping the counties into six districts using the USDA agricultural groupings. We computed land use changes at the district level to account for regional land use changes.. We grouped the counties into six districts based on the USDA grouping system. District 1: Colbert, Franklin, Lauderdale, Lawrence, Limestone, Madison, Marion, Morgan, Winston. District 2: Blount, Calhoun, Cherokee, Cleburne, Cullman, DeKalb, Etowah, Jackson, Marshall, St. Clair. District 3: Bibb, Chambers, Chilton, Clay, Coosa, Fayette, Jefferson, Lamar, Lee, Pickens, Randolph, Shelby, Talladega, Tallapoosa, Tuscaloosa, Walker. District 4: Autauga, Bullock, Dallas, Elmore, Greene, Hale, Lowndes, Macon, Marengo, Montgomery, Perry, Russell, Sumter. District 5: Baldwin, Butler, Choctaw, Clarke, Conecuh, Escambia, Mobile, Monroe, Washington, Wilcox. District 6: Barbour, Coffee, Covington, Crenshaw, Dale, Geneva, Henry, Houston, Pike.

## 3. Results

### *3.1. State Land Use Statistics*

#### 3.1.1. Timberland

We illustrate that about 71% of the total land area are timberlands (Table 2). This makes timberlands the most common land use type in Alabama. From Table 2, timberlands have experienced tremendous changes from 1990–2018. Overall, we indicate that timberlands have increased by about 5%, with the highest increase occurring in 1990–2000 (3.7%) and the lowest increase occurring in 2000–2018 (0.3%). Natural timberlands consist of hardwoods, whereas commercial timberlands are generally monoculture pine interspersed with clear cuts. The increment in timberlands are perhaps due to the establishment of fast-growing softwood plantations over the years [43]. Softwoods have increased about 40% relative to hardwood and mixed hardwood, which have decreased by approximately 3% and 36%, respectively, from 1990–2018. See Table A1 in the appendix section for a detailed summary of timberland statistics. Timberland is important as it provides, on average, $2.2 billion in direct employment earnings annually [44]. In addition, timber industries in Alabama are ranked second in pulp production, paper and paperboard production, and sixth in lumber and panel production across the country [7]. Outdoor recreation adds to the revenue stream, as recreational demands from timberlands generate about $1.8 billion annually [45]. Thus, timberland revenue highlights its importance in the local economy and explains its increasing shares over the years.

#### 3.1.2. Agricultural Land

Agricultural lands are the second most common land use type, constituting about 26% of the total land area (Table 2). Agriculture has significant economic importance, as the region leads in broilers production, as well as other agricultural commodities, such as soybeans, pecans, and cotton [44]. However, our statistics demonstrate that agricultural lands are declining (Table 2). Our analysis shows a 15% decline in agricultural lands from 1990–2018 (Table 2). The highest decline (12%) was experienced between 1990–2000, while the lowest decline (2%) was experienced between 2000–2018. The decline in agricultural lands may be due to the waning economic health of the agricultural sector and over-reliance on timber revenue [5,46]. In addition, environmental programs, such as the CRP, target croplands and encourage landowners to retire sensitive agricultural lands for conservation use. Landowners that enroll in the conservation program are encouraged to plant genetically modified and fast-growing tree species, such as loblolly pine (*Pinus taeda*), for rental fees and cost–sharing benefits. Thus, a decline in crop prices may favor agricultural land conversion for timber plantation.

#### 3.1.3. Urban Land

Urban lands are the third most common land use type. Our analysis shows a 79% increment in urban lands from 1990–2018 (Table 2). We indicate that urban lands have increased more than agricultural and timberlands. The rise in urban lands is due to socio-economic factors, such as population, housing, road construction, and other developmental projects [14] Research shows that urban land values have increased compared to the other land use types. A recent study indicates that urban land prices have increased and range from $7506–$10,322 per acre [5].Future projections suggest that these prices are expected to increase due to the rise in household incomes [2].

**Table 2.** Land Use Changes in Alabama (1000 acres).

| Land Use Types (Acres) | 1990 | 2000 | 2010 | 2018 | % Change (1990–2000) | % Change (2000–2010) | % Change (2010–2018) | % Change (1990–2018) |
|---|---|---|---|---|---|---|---|---|
| Timberland | 21,925.4 (67.4%) | 22,743.2 (70%) | 22,917.7 (70.6%) | 22,997.3 (70.8%) | 3.7 | 0.7 | 0.3 | 4.8 |
| Agriculture | 10,011.5 (30.8%) | 8794.8 (27.1%) | 8591.3 (26.4%) | 8469.7 (26.1%) | −12.2 | −2.3 | −1.4 | −15.4 |
| Urban | 434.9 (1.3%) | 480.9 (1.5%) | 592.9 (1.8%) | 779.5 (2.4%) | 10.6 | 23.3 | 31.5 | 79.2 |
| CRP | 118.9 (0.4%) | 471.8 (1.5%) | 388.8 (1.2%) | 244.2 (0.8%) | 296.7 | −17.6 | −37.2 | 105.4 |
| Total land area | 32,491 | 32,491 | 32,491 | 32,491 | | | | |
| | | | *District Land Use Types* | | | | | |
| **DISTRICT 1** | | | | | | | | |
| Timberland | 1976 (52.5%) | 2058.2 (54.7%) | 2080.4 (55.3%) | 2063.1(54.8%) | 4.1 | 1.08 | −0.8 | 4.4 |
| Agriculture | 1718.5 (45.6%) | 1575.6 (41.8%) | 1550 (41.2%) | 1583 (42%) | −8.3 | −1.6 | 2.1 | −7.9 |
| Urban | 60.7 (1.6%) | 67.5 (1.8%) | 81.2 (2.3%) | 91.2 (2.4%) | 11.1 | 20.5 | 12.3 | 50.3 |
| CRP | 10.0 (0.3%) | 471.8 (1.5%) | 388.8 (1.2%) | 244.2 (0.8%) | 296.7 | −17.6 | −37.2 | 105.4 |
| Total area | 3765.2 | 3765.2 | 3765.2 | 3765.2 | | | | |
| **DISTRICT 2** | | | | | | | | |
| Timberland | 2529.2 (57.6%) | 2650.5 (60.3%) | 2562.7 (58.3%) | 2438 (55.5%) | 4.8 | 3.3 | −4.9 | −3.6 |
| Agriculture | 1777.4 (40.5%) | 1629 (40.5%) | 1721.7 (39.2%) | 1819 (41.4%) | −8.4 | 5.7 | 5.7 | 2.3 |
| Urban | 81.5 (1.9%) | 76.2 (1.9%) | 88.6 (2%) | 117.2 (2.7%) | −6.5 | 16.3 | 32.2 | 43.8 |
| CRP | 4.8 (0.1%) | 37.7 (0.8%) | 19.9 (0.5%) | 18.7 (0.4%) | 685.4 | −47.2 | −6.0 | 291.6 |
| Total area | 4392.9 | 4392.9 | 4392.9 | 4392.9 | | | | |
| **DISTRICT 3** | | | | | | | | |
| Timberland | 5726.6 (75.5%) | 5858.9 (77.2%) | 5848.8 (77.1%) | 5866.3 (77.3%) | 2.3 | −0.2 | 0.3 | 2.4 |
| Agriculture | 1777.7 (23.4%) | 1617.3 (21.3%) | 1630.3 (21.5%) | 1549.7 (20.4%) | −9.0 | 0.8 | −4.9 | −12.8 |
| Urban | 76.5 (1%) | 78.4 (1%) | 89.4 (1.2%) | 155.9 (2.1%) | 2.4 | 14.1 | 74.4 | 103.7 |
| CRP | 6.1 (0.1%) | 32.4 (0.4%) | 17.5 (0.2%) | 15.1 (0.2%) | 429.8 | −46.1 | −13.6 | 146.6 |
| Total area | 7589 | 7589 | 7589 | 7589 | | | | |
| **DISTRICT 4** | | | | | | | | |
| Timberland | 3871.5 (63.9%) | 4253.6 (70.2%) | 4325.6 (71.3%) | 4402 (72.6%) | 9.9 | 1.7 | 1.8 | 13.7 |
| Agriculture | 2023.9 (33.2%) | 1536 (25.3%) | 1479.1 (24.4%) | 1375 (22.7%) | −24.1 | −24.1 | −3.7 | −32.1 |
| Urban | 102.2 (1.7%) | 125.1 (2.1%) | 144.8 (2.4%) | 205.3 (3.4%) | 22.4 | 22.4 | 15.7 | 100.8 |
| CRP | 65.3 (0.1%) | 148.7 (2.5%) | 113.8 (1.9%) | 80.5 (1.3%) | 127.6 | 127.6 | −23.4 | 23.2 |
| Total area | 6062.9 | 6062.9 | 6062.9 | 6062.9 | | | | |
| **DISTRICT 5** | | | | | | | | |
| Timberland | 5316 (78.9%) | 5223.1 (77.5%) | 5424.3 (80.5%) | 5521 (81.9%) | −1.7 | 3.8 | 1.8 | 3.9 |
| Agriculture | 1369.1 (20.3%) | 1416.8 (21%) | 1218.8 (18.1%) | 1141.8 (16.9%) | 3.5 | −14.0 | −6.3 | −16.6 |
| Urban | 43.9 (0.7%) | 47.6 (0.7%) | 47.7 (0.7%) | 56.4 (0.8%) | 8.3 | 0.2 | 18.5 | 28.6 |
| CRP | 12.8 (0.2%) | 53.6 (0.8%) | 51 (0.8%) | 22.4 (0.3%) | 320.3 | −5.0 | −51.6 | 75.7 |
| Total area | 6741.8 | 6741.8 | 6741.8 | 6741.8 | | | | |
| **DISTRICT 6** | | | | | | | | |
| Timberland | 2506.1 (63.6%) | 2698.3 (68.5%) | 2691.4 (68.3%) | 2708 (68.9%) | 7.7 | −0.3 | 0.6 | 8 |
| Agriculture | 1344.9 (34.1%) | 1021 (25.9%) | 991.3 (25.2%) | 1016 (25.8%) | −24.1 | −2.9 | 2.5 | −24.4 |
| Urban | 70.0 (1.8%) | 86.2 (2.2%) | 117.1 (3%) | 139.6 (3.5%) | 23.1 | 35.8 | 16.4 | 94.6 |
| CRP | 19.8 (0.5%) | 135.4 (3.4%) | 141.1 (3.6%) | 77.4 (2%) | 582.4 | 3.5 | −44.7 | 290.3 |
| Total area | 3940.9 | 3940.9 | 3940.9 | 3940.9 | | | | |

Note: Land use percentages are denoted in parenthesis.

### 3.1.4. Conservation Reserve Program Land

The CRP is an environmental initiative that retires sensitive agricultural lands from active production for conservation use [47]. Our analysis indicates that CRP lands are the scarcest land-use type. We demonstrate that the largest enrollment occurred in 1990–2000 (Table 2). However, there has been a decline in enrollment over the years (Table 2). The enrollment dynamics can be ascribed to the nature of the CRP contract. The CRP is a 'lock-in' program and landowners that enroll in the program can have access to their lands for agricultural use after 12–14 years. During the contract period, landowners are expected to adopt prescribed land management practices. Comparing CRP and agricultural lands,

we reveal that landowners enrolled vast acreages of agricultural lands when the program began. However, these original agricultural lands were mostly used for pine plantations. This explains the increase in timberlands (Table 2).

### 3.2. District Land Use Statistics

#### 3.2.1. Timberland

Timberlands have increased in all the districts, except district two, which shows a 4% decline from 1990–2018 (Table 2). The remaining five districts have increased in timberland areas in varying proportions. For instance, districts three and four's timberland areas have increased by about 2% and 14%, respectively (Table 2). Hardwoods have declined in all districts except for districts two and four. Districts one and five's timberland areas have increased by about 4% each, while district six's has increased by about 8% (Table 2). These changes emanate from softwood establishments (see Table A1). The increase in softwoods in districts three, four, five, and six are expected, as these places are the hub of the forest product industries and account for the majority of the region's timber revenue [7]. Timberland owners in districts five and six generate revenue from timber harvesting, while owners in districts one and two obtain revenue from timberland recreational use [5]. The non-timber revenue obtained by timberland owners in districts one and two accounts for the retention of hardwoods, thereby the increase in hardwood areas.

#### 3.2.2. Agricultural Land

The agricultural land use statistics show that district two gained about a 3% increase from 1990–2018 relative to the other districts (Table 2). The rise in agricultural lands in district two is consistent with cropland values, as counties in north Alabama have higher cropland values compared with those in south Alabama [5]. Agricultural lands have declined in the districts, and district four has experienced the highest decline of 32%. Districts one, three, five, and six have decreased in agricultural lands by about 8%, 13%, 17%, and 25%, respectively.

#### 3.2.3. Urban Land

There has been an increase in urban lands across all the districts due to urbanization and population growth [48]. For instance, urban lands in districts three and four have appreciated compared with the other districts. This is because districts three and four are in the Piedmont regions, and these areas have experienced high population growth over the years [4]. The rise is also due to industrialization, the closeness of some districts to major metropolitan areas, and easy access to highways [2,4].

#### 3.2.4. Conservation Reserve Program Land

The statistics indicate that districts four, five, and six have many acres of CRP land compared with districts one, two, and three. Table 2 shows a decline in CRP land across all districts from 2010–2018. The decline in CRP lands is prevalent in north Alabama compared with south Alabama. This may be due to the conversion of CRP lands into timberlands, as CRP management practices encourage tree planting as a conservation measure.

### 3.3. Results and Discussions

Table 3 presents the results and test statistics for the econometric analysis. Our results indicate a good model fit and are significant with intuitive interpretations. The R-squared values range from 0.41 to 0.64 for the various land use types. This indicates a good prediction of our model. The values of (prob > chi2) are <0.001 in all four land use categories determine that our results are significant in explaining variations in the land use types. Table 3 gives the percentage change in land use that results from a percentage change in an explanatory variable, with other explanatory variables held constant.

**Table 3.** Parameter Estimates [1] of PSUR Model.

| Independent Variables [2] | Dependent Variables | | | |
|---|---|---|---|---|
| | Timberland | Agricultural Land | Urban Land | CRP Land |
| Population density | −0.1598 *** | 0.4338 *** | 0.2421 *** | 0.0365 |
| | (0.0077) | (0.1777) | (0.0452) | (0.0829) |
| Income per capita | 0.0951 *** | −0.3736 *** | 0.3446 *** | −0.2967 *** |
| | (0.0149) | (0.0338) | (0.0877) | (0.1595) |
| Land quality I–II | 0.0135 | 0.0464 *** | −0.1549 ** | −0.1193 |
| | (0.0124) | (0.0285) | (0.0732) | (0.1195) |
| Land quality III–IV | 0.0147 | 0.1217 *** | 0.4921 *** | 1.8422 *** |
| | (0.0259) | (0.0558) | (0.1505) | (0.2758) |
| Land quality V–VIII | 0.1108 *** | −0.0429 *** | 0.8745 *** | 1.33 *** |
| | (0.0234) | (0.0512) | (0.1321) | (0.2432) |
| Crop price index | −0.0537 *** | 0.0437 *** | −1.257 *** | −2.0113 *** |
| | (0.016) | (0.0148) | (0.0256) | (0.675) |
| Timber price | 0.0745 *** | −0.1847 *** | −0.6584 *** | 1.3517 *** |
| | (0.0141) | (0.0308) | (0.0807) | (0.1384) |
| Interest rate | −0.0121 *** | 0.0235 *** | 0.0083 | −0.3754 *** |
| | (0.0032) | (0.007) | (0.0189) | (0.0336) |
| Disaster | 0.0522 *** | 0.018 | 0.0376 | 0.2837 |
| | (0.0188) | (0.0439) | (0.1095) | (0.2041) |
| Root MSE | 0.1417 | 0.2949 | 0.4709 | 0.9298 |
| R-square | 0.6467 | 0.6649 | 0.4033 | 0.5761 |
| Chi2 | 431.98 | 468.36 | 159.53 | 320.75 |
| *p value* | <0.0001 | <0.0001 | <0.0001 | <0.0001 |

Note: [1] Figures in parenthesis are the standard errors. *** $p < 0.01$; ** $p < 0.05$. [2] All explanatory variables are in logarithm terms except the interest rate and the disaster variables.

Our estimates suggest that high quality lands are allocated for agricultural use but not for timberland purposes. Again, our findings support the notion that higher net returns for particular land use are positively associated with own use but have a negative effect on alternative uses. For instance, we reveal that timber prices encourage converting land for timber production, while crop prices support the usage of land for agriculture. Our findings align with past studies which suggest that higher net returns for a particular use favor land conversion to that use [1,10,13]. We show that a 1% increase in timber price is associated with a 0.07% increase in timberlands, but a 0.18% and 0.66% decrease in agricultural and urban lands. Again, we find that a 1% increase in timber price leads to a 1.35% increase in CRP land. We expected the relationship between timber prices and CRP lands, as most CRP lands are mainly for tree pine plantations. A recent study indicates that CRP induces a significant increase in the supply of sawtimber, as rental payments reduce plantation costs [27]. We also show that the crop price index has a positive and significant effect on agricultural lands, as a 1% increase in crop prices is associated with a 0.05% increase in agricultural lands. However, crop prices have a negative effect on timberlands, urban lands, and CRP lands (Table 3). We expect these outcomes, as previous studies reveal a negative relationship between crop prices and CRP lands [49–51]. However, the relationship between crop prices and urban lands is unexpected as our framework suggests a positive relationship. Nevertheless, our results align with other past studies [13]. Furthermore, we highlight that urban lands decline with increasing timber prices. This is important and has implications for the housing market as lower (higher) timber prices can translate into less (more) expensive wood products for construction.

Our results support the hypothesis that timberlands and agricultural lands compete with each other. We reveal that a 1% increase in timber prices causes a 0.18% decrease in agricultural lands, whereas a 1% increase in crop prices causes a 0.05% decrease in timberlands. Cross-prices explain how land use changes are affected by commodity prices and also serve as a tool to investigate potential land use externalities. Subsequently, we show that interest rates affect timberlands and urban lands negatively but exhibit a positive

effect on agricultural lands. These findings emphasize that lower interest rates favor timberlands and urban lands, as these land types require higher capital investment. When interest rates increase, we expect investments in timberland and urban lands to decline due to an increase in operational costs. We evaluate the effect of natural disasters on land use and show those counties that suffer from natural disasters are likely to establish timberlands, whereas counties that are not disaster-prone are unresponsive to agricultural lands, urban lands, and CRP lands.

To determine the effect of land quality on land use change, we reveal that high quality lands have an insignificant effect on timberlands and CRP lands but are positively and significantly related to agricultural lands. These results are intuitive, as lands with good quality will not be allotted for conservation use. From Table 3, we show that low quality lands have a positive effect on timberland, urban lands, and CRP lands, but are negatively related to agricultural land. These results conform to a prior expectation of previous studies that low quality lands increase the per unit cost of intensive land use activities [10,13]. Again, we prove that landowners find it beneficial to allocate low quality lands for conservation purposes in exchange for rental payments and benefits. Turning to medium land quality, we show these soil types have a positive and significant effect on all the land use types except timberlands. This soil type supports agriculture in pasture use. Pasture is an essential component of agriculture in Alabama through livestock production [4]. Also, we notice from our results that land conversion from agriculture to timber is evident when land declines in quality. We confirm that agricultural lands are more competitive than timberlands when the quality of land is high.

We turn our attention to the socioeconomic factors and reveal that urban and agricultural lands are an increasing function of the population. We show that a 1% increase in the population density is associated with a 0.43% and 0.24% increase in agricultural and urban lands, respectively, but a 0.15% decrease in timberlands. These results confirm that increasing population causes urban expansion and promotes the establishment of developed areas [12,14]. Linking this outcome to tropical deforestation, we realize that population growth increases the conversion of tropical forest and timberland for developmental purposes [15,16,52]. As expected, we demonstrate that increasing population density drives the demand for land for agricultural purposes.

Furthermore, we establish that household income is an increasing function of urban lands and timberlands, but not agricultural and CRP lands. We find that a 1% increase in the median household income per capita is associated with a 0.09% and a 0.34% increase in timberlands and urban lands, respectively. However, the household income decreases agricultural and CRP lands by 0.37% and 0.29%, respectively. Comparing the coefficients of the land use types, we find that the effect is magnified in urban land use. This is expected and consistent with the literature, as income plays a major role in urbanization and determines recreational demand from timberland [1,41].

## 4. Conclusions and Policy Implications

We analyzed factors that influence land use change in Alabama. We adopted county-level data and employed a method that mitigated the endogeneity and correlation issues among the variables. We explored the cross-section and time-variations among the land use types to estimate the parameters which we utilized to recommend policies in regard to land use changes.

We indicate that timberlands are the predominant land use type and constitute about 71% of the total land use. Timberland areas are mainly occupied by softwood species and constitute about 65% of the total timberland mix. Agricultural lands are the second most dominant land use type and constitute about 26% of the total land area. Urban lands rank third, and the scarcest land-use type is CRP lands. We show that urban lands and timberlands have appreciated, whereas agricultural and CRP lands have declined. Turning to the district land use changes, we show that timberland areas have increased in all districts except district two. Also, the district agricultural land statistics indicate that,

apart from district two, which has experienced an increase in agricultural lands, all other districts have experienced a decline in agricultural lands. We show that urban lands have increased in all the districts. As part of this study, we indicate that districts in southern Alabama have more CRP lands compared to the other districts. Thus, we propose that areas experiencing decline in CRP lands should be given high priority for the sake of biodiversity and environmental service protection.

Our estimates from the system of equations have good statistical validity, are statistically significant, and meet expected outcomes. Generally, we demonstrate that land use changes are due to socio-economic pressures, market structure, land quality, and natural factors. Our estimates indicate the role of economic gains in spurring land conversion to alternative uses and the important effect of land quality in land allocation use decisions. The estimates suggest that prices of own-use retain land for that particular use and influence the conversion to that particular use. Our analysis suggest that socioeconomic factors affect land demand to meet specific needs as population growth causes timberland decline but is a spurring agent for urban land use. We indicate that timberlands tend to be more responsive to timber prices, household income, and poor soil quality, while agricultural lands positively respond to lands with good soil quality, crop prices, interest rates, and population growth. We show that household income per capita drives urban land for residential, industrial and infrastructure purposes. Again, increasing household income encourages timberland establishments. Capital cost is also essential in increasing timberlands. These findings are important market-based tools that policy-makers and government agencies can utilize to influence land-use changes through deliberate policy interventions.

Our findings have several policy implications. First, the combination of the endogeneity correction method with the system of equations have proven to be an effective and reliable technique to determine the drivers of land use change. We recommend that this approach be adopted and applied to explore similar topics that seek to evaluate land use changes. While timber prices were sensitive to CRP lands, we suggest that incentive-based policies that increase returns for holding CRP lands for environmental purposes be encouraged. Moreover, we recommend policies that encourage establishing timberland to achieve environmental sustainability. Similarly, policies geared towards the conservation use of land must incorporate measures to prevent the conversion of such lands for agriculture in the wake of high crop prices, as empirical evidence suggest that payments offered to farmers for conservation encourage participation when crop prices are low, but the reverse is observed when crop prices are high [51]. Thus, it is imperative to formulate policies that prevents switching conservation lands for agriculture to sustain the environmental and economic benefits that are derived from CRP lands.

Although we adopt a novel approach to evaluate land use changes, there are limitations that need to be addressed in future studies. One caveat in our study is the limited and unavailable county-level land conversion cost data. Owing to this limitation, we are forced to assume that land conversion cost is implicitly embedded in the land quality classes following past studies [10,13]. Although we try to correct for the endogeneity issues by using lag values of population density and household income for urban land use, researchers indicate that using lagged values as instruments are not very effective for solving such econometric problems [53]. Future studies should find potential IVs to resolve these issues. Lastly, future studies can adopt nonlinear models to evaluate the thresholds that landowners are likely to respond to the determinants of land use change.

**Author Contributions:** Conceptualization, E.A. and Y.Z.; methodology, E.A.; software, E.A.; validation, E.A., Y.Z., W.L. and L.N.; formal analysis, E.A., Y.Z., W.L. and L.N.; investigation, E.A.; resources, E.A.; data curation, E.A.; writing—original draft preparation, E.A.; writing—review and editing, E.A., Y.Z., W.L. and L.N.; supervision, Y.Z.; project administration, Y.Z. All authors have read and agreed to the published version of the manuscript.

**Funding:** This research received no external funding.

**Data Availability Statement:** Data are already in published manuscripts or can be obtained upon request.

**Conflicts of Interest:** The authors declare no conflict of interest.

## Appendix A

**Table A1.** District level timberland statistics.

| State/District/ | 1990 | 2000 | 2010 | 2018 | % Change | % Change | % Change | % Change |
|---|---|---|---|---|---|---|---|---|
| Land use types | | | | | (1990–2000) | (2000–2010) | (2010–2018) | (1990–2018) |
| Alabama Timberland Use Types | | | | | | | | |
| Hardwood | 9947.3 | 10,543.4 | 10,095.2 | 9674.8 | 6.0 | −4.3 | −4.2 | −2.8 |
| | 45.37% | 46.36% | 44.05% | 42.07% | | | | |
| Softwood | 7456.6 | 8006.1 | 9308.8 | 10,407.2 | 7.4 | 16.3 | 11.8 | 39.6 |
| | 34.01% | 35.20% | 40.62% | 45.25% | | | | |
| Mixed Hardwood | 4521.5 | 4193.7 | 3513.7 | 2915.3 | −7.4 | −16.2 | −17 | −35.5 |
| | 20.62% | 18.44% | 15.33% | 12.68% | | | | |
| Total area | 21,925.4 | 22,743.2 | 22,917.7 | 22,997.3 | | | | |
| District Timberland Use Types | | | | | | | | |
| DISTRICT 1 | | | | | | | | |
| Hardwood | 1272.9 | 1299.7 | 1272.7 | 1249 | 2.1 | −2.1 | −1.9 | −1.9 |
| | 64.42% | 63.15% | 61.18% | 60.57% | | | | |
| Softwood | 430.4 | 431.9 | 537.9 | 591 | 0.3 | 21 | 13.1 | 37.3 |
| | 21.78% | 20.98% | 25.86% | 28.66% | | | | |
| Mixed hardwood | 272.7 | 326.6 | 269.8 | 222 | 19.8 | −17.4 | −17.7 | −18.6 |
| | 13.80% | 15.87% | 12.97% | 10.77% | | | | |
| Total area | 1976 | 2058.2 | 2080.4 | 2062 | | | | |
| DISTRICT 2 | | | | | | | | |
| Hardwood | 1360.1 | 1555 | 1456.4 | 1364 | 14.3 | −6.3 | −6.3 | 0.3 |
| | 53.78% | 58.67% | 56.83% | 55.95% | | | | |
| Softwood | 653.7 | 568.8 | 659.9 | 707 | −13 | 16 | 7.1 | 8.1 |
| | 25.85% | 21.46% | 25.75% | 29.00% | | | | |
| Mixed Hardwood | 515.4 | 526.8 | 446.5 | 367 | 2.2 | −15.2 | −17.8 | −28.8 |
| | 20.38% | 19.87% | 17.42% | 15.05% | | | | |
| Total area | 2529.2 | 2650.6 | 2562.8 | 2438 | | | | |
| DISTRICT 3 | | | | | | | | |
| Hardwood | 2659.5 | 2559.8 | 2416.7 | 2293.5 | −3.7 | −5.6 | −5.1 | −13.8 |
| | 46.44% | 43.69% | 41.32% | 39.10% | | | | |
| Softwood | 1742.7 | 2079.3 | 2415.6 | 2736.1 | 19.3 | 16.2 | 13.2 | 57 |
| | 30.43% | 35.49% | 41.30% | 46.64% | | | | |
| Mixed Hardwood | 1324.4 | 1219.8 | 1016.5 | 836.6 | −7.9 | −16.7 | −17.7 | −36.8 |
| | 23.13% | 20.82% | 17.38% | 14.26% | | | | |
| Total area | 5726.6 | 5858.9 | 5848.8 | 5866.2 | | | | |
| DISTRICT 4 | | | | | | | | |
| Hardwood | 1676.1 | 1984 | 1958 | 1931.5 | 18.3 | −1.3 | −1.3 | 15.2 |
| | 43.29% | 46.64% | 45.27% | 43.88% | | | | |
| Softwood | 1392.2 | 1557 | 1787.5 | 2001.5 | 11.8 | 14.8 | 11.9 | 43.8 |
| | 35.96% | 36.60% | 41.33% | 45.47% | | | | |
| Mixed Hardwood | 803.2 | 713 | 578 | 469 | −24.1 | −3.7 | −7 | −32.1 |
| | 20.75% | 16.76% | 13.40% | 10.65% | | | | |
| Total area | 3872 | 4254 | 4325 | 4402 | | | | |
| DISTRICT 5 | | | | | | | | |
| Hardwood | 1836.3 | 1930.1 | 1834.2 | 1733.8 | 5.1 | −5 | −5.5 | −5.6 |
| | 34.54% | 36.95% | 33.81% | 31.40% | | | | |
| Softwood | 2359.2 | 2325.2 | 2777.2 | 3107.7 | −1.4 | 19.4 | 11.9 | 31.7 |
| | 44.38% | 44.51% | 51.20% | 56.29% | | | | |
| Mixed Hardwood | 1120.5 | 968.4 | 813 | 679.4 | −13.6 | −16 | −16.4 | −39.4 |
| | 21.08% | 18.54% | 14.99% | 12.31% | | | | |
| Total area | 5316 | 5223.7 | 5424.4 | 5520.9 | | | | |
| DISTRICT 6 | | | | | | | | |
| Hardwood | 1142.4 | 1215 | 1157.3 | 1102.6 | 6.4 | −4.8 | −4.7 | −3.5 |
| | 45.58% | 45.03% | 43.00% | 40.72% | | | | |
| Softwood | 878.4 | 1044 | 1145.9 | 1263.7 | 18.8 | 9.8 | 10.3 | 43.9 |
| | 35.05% | 38.69% | 42.57% | 46.67% | | | | |
| Mixed Hardwood | 485.3 | 439.1 | 388.3 | 341.4 | −9.5 | −11.6 | −12.1 | −29.7 |
| | 19.36% | 16.27% | 14.43% | 12.61% | | | | |
| Total area | 2506.1 | 2698.1 | 2691.5 | 2707.7 | | | | |

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
