# Peer review of "What Drives Land Use Change in the Southern U.S.? A Case Study of Alabama"

_forests, doi:10.3390/f14020171_

Round 1
Reviewer 1 Report
Thanks for inviting me to review this paper. My comments are as follows.
The research focuses on the determination of the land use change in the Southern US. An important question that remains to be clearly answered is why it is important? And why studying this issue in a particular location in Alabama, US should be important to general readers? The current motivation is insufficient and should be carefully improvement.
Literature review is currently weak. The authors should clearly discuss how the research connects with the existing literature in detail. The discussion should not just simply state what the existing research has found but summarize whether and how your research is different from existing ones?
When reporting the empirical results, the authors should explain whether and how their obtained research are in line with the existing findings. In addition to introducing what you have found, it is always more important to tell the reader how to interpret the same.
The current empirical analysis is too simple to convince its reliability to the readers. For example, how does the current research deal with any potential endogeneity issue including simultaneity, reverse causality, etc.?
The analysis involves several types of land use changes. The corresponding research implications should be highlighted further, and not just a simple and general narrative.
Reviewer 2 Report
Comments on paper forests-2022128
1.This paper presents a comprehensive study on the drives factors of land use change in the Southern U.S.A. The study serves as a good example for how to Judge the drives factors of land use change, and which is helpful for understanding the drives factors of land use change. I believe this paper is a good one for the forests.
2.However, the following comments should be useful to polish the quality of the presentation of the paper from the reader's benefit:
(1)The introduction is not clear enough. I recommend rewrite abstract and introduction.
(2) I believe the authors had collected various data sets for the research. But the data sources of different data are not clearly stated, so add the data sources clearly.
(3) the main research aim of the paper is to discuss the drivers of the land use change in the southern U.S. but you only considered the social and economic factors , natural factors are not considered. So, I suggest that the global climate change would be into considerations.
(4)And In the discussion section you have to discuss the differences between your research and previous research.
Author Response
Please, see the attachment

Reviewer 3 Report
Reviewer’s comments:
I have some comments on your paper titled “What drives land use change in the Southern U.S.? A Case Study of Alabama, USA”. Overall, your paper topic is quite interesting and this work has a potential to contribute to the literature on land management and sustainability. However, many parts of the paper are still weak. Hence, author(s) are required to carefully revise to strengthen your work.
Specific comments:
1. Please elaborate on the Method Section in the revised manuscript. For example, please provide more information on how you built your model, why you chose the model’s variables, what tests you ran to validate your model before interpretation, and so on. Please also provide more information on your model, as shown in Table 3.
2. The Discussion Section is still weak. Please provide and/or elaborate as much as possible on this section. How do the findings relate to previous findings of many previous works in places around the world such as Vietnam, Indonesia, and so on...
3. Please provide and/or further elaborate on the Limitation Section of your paper.
4. Some key references that you may read and use for your revising:
· Call, M.; Mayer, T.; Sellers, S.; Ebanks, D.; Bertalan, M.; Nebie, E.; Gray, C. Socio-Environmental Drivers of Forest Change in Rural Uganda. Land use policy 2017, 62, 49–58, doi:10.1016/j.landusepol.2016.12.012.
· Hosonuma, N.; Herold, M.; De Sy, V.; De Fries, R.S.; Brockhaus, M.; Verchot, L.; Angelsen, A.; Romijn, E. An Assessment of Deforestation and Forest Degradation Drivers in Developing Countries. Environ. Res. Lett. 2012, 7, doi:10.1088/1748-9326/7/4/044009.
· Khuc, Q. Van; Tran, B.Q.; Meyfroidt, P.; Paschke, M.W. Drivers of Deforestation and Forest Degradation in Vietnam: An Exploratory Analysis at the National Level. For. Policy Econ. 2018, 90, 128–141, doi:10.1016/j.forpol.2018.02.004.
· Van Khuc, Q.; Le, T.A.T.; Nguyen, T.H.; Nong, D.; Tran, B.Q.; Meyfroidt, P.; Tran, T.; Duong, P.B.; Nguyen, T.T.; Tran, T.; et al. Forest Cover Change, Households’ Livelihoods, Trade-Offs, and Constraints Associated with Plantation Forests in Poor Upland-Rural Landscapes: Evidence from North Central Vietnam. Forests 2020, 11, doi:10.3390/F11050548.
· Meyfroidt, P.; Roy Chowdhury, R.; de Bremond, A.; Ellis, E.C.; Erb, K.H.; Filatova, T.; Garrett, R.D.; Grove, J.M.; Heinimann, A.; Kuemmerle, T.; et al. Middle-Range Theories of Land System Change. Glob. Environ. Chang. 2018, 53, 52–67, doi:10.1016/j.gloenvcha.2018.08.006.
Author Response
Please, see the attachment

Round 2
Reviewer 3 Report
Thank you for your appropriately addressing all of my comments and the updated version is now much improved. I have no further comment on your work.
Author Response
Dear Reviewer,
Thank you very much for your review.